# OpenReview forum: "Test-Time Learning of Causal Structure from Interventional Data"
_ICLR.cc/2025/Conference — Submitted to ICLR 2025_

### Official Review · Reviewer_8DQE · 2024-11-03

**Soundness:** 2
**Presentation:** 2
**Contribution:** 2
**Rating:** 5
**Confidence:** 4

**Summary:**

This paper introduces a supervised causal discovery framework that is composed of three steps. First, they convert the given interventional data collection into an augmented observational data samples. Then, they generate simulation training data from the augmented samples. Finally, they tune a supervised PC algorithm for identifiying the causal graph.

**Strengths:**

**1**. The paper is clearly written and easy to read.

**2**. The authors conduct extensive experiments to validate their method.

**Weaknesses:**

**1.** The novelty of this paper is limited. The first step, pooling interventional data into observational data samples, was introduced by Joint Causal Inference (Moiij et al., 2020). The second step, generating simulated trainning data via a proxy algorithm and Maximum Likelihood Estimation (MLE) for estimating the conditional probability, has been proposed by ML4S (Sec. 5; Ma et al., 2022). This paper is just a trivial combination of existing methods.

**2**. The supervised PC algorithm (Step 3) is problematic. Now that you already test conditional independence from data (Sec. 4.1, Freaturization step), why do you use a supervised classifier to determine the v-structure, instead of adopting the orientation rules that enjoy the theoretical identifiability? The approach costs more computations (to train the classifier) and meanwhile loses the identificiation guarantee.

**Questions:**

See the weakness above.

---

> ### Author Response · Authors · 2024-11-25
> **The First Part of Response to Reviewer 8DQE**
>
> Dear Reviewer 8DQE,
>
> We sincerely appreciate your valuable time and the effort you have dedicated to reviewing our paper. We are also pleased that you recognize several strengths of our work, including its ***clarity*** and ***extensive experiments***. However, we regret that you have raised a criticism that we believe to be _inappropriate_. **We have carefully considered each of your comments and addressed them systematically to respond to your concerns**. Specifically:
>
> ---
>
> > **W1: Limited novelty, Step 1 is JCI, Step 2 is ML4S**
>
> - _We **respectfully disagree with your assessment**. We must emphasize that **this is not a mere incremental contribution**, **but rather the result of long-term, rigorous work**_.
>     - Both reviewers, **4MTP** and **dQ5H**, pointed out that our paper presents a _**novel paradigm**_. In particular, reviewer **dQ5H** highlighted that our approach combines two existing techniques (JCI and SCL), providing a _**well-described and interesting combination that is both intuitive and clever**_. We would like to emphasize that _**the presence of similar existing concepts does not negate the novelty**_ of our work. Rather, it is _**the ability to effectively address the problem**_ at hand, as well as _**confronting the challenges**_ throughout the process, that _**constitutes the true source of innovation**_. Furthermore, given the complexity of the problem we are tackling, _**even applying existing concepts requires significant and non-trivial effort**_.
>     - Based on the feedback from all reviewers, _we realize that we have presented many things, which may have led to the core novelty of our work not being immediately clear to the readers_.
>     - Given the complexity and importance of the problem, we have _**rewritten the introduction** to systematically clarify what we are doing and how to identify the unique contributions of our work_. This restructured introduction now more clearly outlines our key contribution from the perspectives of **problem**, **idea**, and **technology** to aid reader understanding. Please also refer to our [_**"General Response to All Reviewers Regarding the Main Contributions of Our Paper"**_](https://openreview.net/forum?id=ZXs3pkmrRG&noteId=Qq23h2Ckyd).
>     - To briefly summarize, our core contribution lies in utilizing _**Supervised Causal Learning (SCL)** tools_ to address _**Interventional Causal Discovery (ICD)** problem_, for which we propose the novel _**Test-Time Training (TTT) + Joint Causal Inference (JCI)** paradigm_ and explore _customized **IS-MCMC**_ and _**PC-Like SCL** algorithms_ for training data acquisition and model learning.
>
> - Here, we would like to specifically address why _**our paper is not simply a combination of JCI and ML4S**_. The key differences include:
>     - **Difference from JCI**: While we acknowledge and appreciate the contributions of JCI, our systematic investigation shows that it is a promising direction for intervention causal discovery, as it simplifies the causal discovery problem and flexibly handles various intervention settings. _**However, JCI does not address key questions** such as: What are the appropriate learning objectives? How should the learning process be designed?_ These are precisely the questions we systematically answer in our paper.
>     - **Difference from ML4S**: _**First, from the problem perspective**_, we note that ML4S focuses exclusively on skeleton recovery, so _**our problem space and difficulty far exceed that of ML4S**_.  _**Second, from the conceptual perspective**_, ML4S’s Vicinal Graph as training data is merely a heuristic method. It does  _**not systematically investigate when to acquire training data, what kind of training data to collect, and how to gather this data under intervention settings.**_ In contrast, we conduct a thorough investigation into acquiring training data for intervention causal discovery. Finally, _**from the technical perspective**_, ML4S uses a simple heuristic algorithm for inference and does not design a rigorous algorithm for data acquisition. _**Our IS-MCMC algorithm can be seen as a generalization of ML4S**_, as reflected in the results of our training data study in the experimental section.
>
> We hope that this detailed explanation helps you truly understand the core idea and significant contributions of our paper, and appreciate the considerable effort we have invested in this long-term work.

---

> ### Author Response · Authors · 2024-11-26
> **The Second Part of Response to Reviewer 8DQE**
>
> ---
>
> > **W2: If conditional independence information is already available, why not use it directly, as this might otherwise undermine theoretical identifiability?**
>
> - In response, we clarify that our _**PC-Liked SCL**_ algorithm retains theoretical identifiability. Moreover, it ensures the correctness of asymptotic properties while encouraging more systematic feature characterization and learning better classification mechanisms, thus making our method more robust than PC. As a result, it achieves superior empirical performance.
> - Additionally, in the revised version, **Appendix D** presents a systematic study of the advantages of our approach over the PC algorithm, from intuitive analysis to theoretical proof. Please also refer to our [_**"General Response to All Reviewers Regarding the Identifiability of Our PC-like SCL Algorithm"**_](https://openreview.net/forum?id=ZXs3pkmrRG&noteId=JzO7AtIHvC).
>
> ---
>
> **We greatly value your feedback**, and we will incorporate these discussions into the final revision of the paper. **We hope that the above responses address your concerns**. **If possible, we kindly request that you reconsider your score.** Should you have any further **constructive suggestions**, we would be happy to discuss them and make necessary improvements to the paper.
>
> Sincerely,
>
> Authors of submission 3081

---

> > ### Comment · Reviewer_8DQE · 2024-11-28
> >
> > I thank the authors for the response. They have addressed my concern regarding the identifiability of the supervised PC algorithm. I will increase my score to 5.

---

> > > ### Author Response · Authors · 2024-11-29
> > > **Thank you for your prompt reply and recognition.**
> > >
> > > Thank you for your prompt feedback and for acknowledging our efforts to address your concern regarding identifiability. We really appreciate your updated score.
> > >
> > > Sincerely,
> > >
> > > Authors of submission 3081

---

### Official Review · Reviewer_4MTP · 2024-11-03

**Soundness:** 3
**Presentation:** 3
**Contribution:** 2
**Rating:** 6
**Confidence:** 3

**Summary:**

The authors tackle the problem of inferring causal structures from interventional data with unknown targets, where standard Supervised Causal Learning (SCL) often fails due to distribution shifts. They introduce TICL (Test-time Interventional Causal Learning), which leverages self-augmentation at test time to adapt the model to test data biases. TICL integrates the Joint Causal Inference (JCI) framework, transforming rule-based logic into a learning-based approach that better utilizes augmented data. Extensive experiments show TICL's effectiveness in causal discovery and target detection.

**Strengths:**

The main contribution of the authors is their proposal of a novel JCI+SCL paradigm for causal discovery under unknown interventions from discrete data. Their TICL approach significantly advances the state of the art in discovering causal relations and interventional targets, achieving an average F1-score improvement across diverse real-world datasets. The authors introduce self-augmentation within SCL, enhancing training instances from the test data, and implement the IS-MCMC algorithm, which effectively samples graphs from the posterior distribution and generates compatible data for training. Their systematic study highlights JCI as a promising direction for interventional causal discovery, with TICL offering flexibility and requiring fewer assumptions in various intervention settings.

**Weaknesses:**

Although the paper is a well-contributed effort on empirical and experimental sides, showing significant advancements over the state of the art, I still believe the paper can improve with some theoretical justification on key steps involved in the proposed approach. For instance, the claim that the algorithm IS-MCMC samples $G$ from the posterior estimation $P(G|D)$ and that the data $D_i$ is compatible with $G_i$ is made. However, the authors do not justify how the stationary distribution of the constructed Markov chain matches the posterior distribution of graphs given the data. Similarly, there is a lack of discussion on the sampling convexity for convergence of the Markov chain. If the authors can clarify these issues, I would be willing to revise my score for the paper and see it as a more valuable contribution to the field.

**Questions:**

I have few question for the authors:

* Can the authors provide some justification on how the stationary distribution of the constructed Markov chain(s) in IS-MCMC matches the posterior distribution of graphs given the data?

* What is a good criterion (or number of samples) to ensure proper convergence for the constructed Markov chain(s)?

* The authors claim that a binary classifier enables us to replace heuristic searches and potentially erroneous conditional independence (CI) tests with a robust classification mechanism. Do the authors have any empirical evidence for this claim in their set of experiments?

* In experiment section 5.1, the authors mention that they sample, or forward-sample, 10,000 samples for both observational cases and each intervention. I wanted to ask how one would choose this number. Is there a risk of overfitting if the chosen sample size is too large? If not, can you explain why it is generally better to use a larger number of samples?

---

> ### Author Response · Authors · 2024-11-25
> **The First Part of the Response to Reviewer 4MTP**
>
> Dear Reviewer 4MTP,
>
> We sincerely appreciate the time and effort you have dedicated to providing insightful feedback on our paper. We are also pleased to see that you recognize several key aspects of our work in the summary and strengths section, including ***extensive experiments***, ***novel paradigm***, ***significant advancement of the state-of-the-art***, ***systematic study***, and ***flexibility***. We have carefully considered each of your detailed comments and addressed them case by case to alleviate your concerns. Specifically:
>
> ---
>
> > **Q1: Why does the stationary distribution of the constructed Markov chain(s) in IS-MCMC matches the posterior distribution of graphs given the data? Is there a discussion of the convexity of the Markov chain? Please provide some justification.**
>
> - As a quick response, we would like to clarify that our IS-MCMC algorithm follows the standard Structure-MCMC procedure in the intervention-augmented graph space. By employing a Metropolis-Hastings sampler, the Markov chain is guaranteed to have a stationary distribution equal to the posterior distribution $P(G | D)$, which has been well studied in the literature[1, 2, 3].
> - For a more detailed discussion, please refer to our [_**"General Response to All Reviewers Regarding the Main Contributions of Our Paper"**_](https://openreview.net/forum?id=ZXs3pkmrRG&noteId=Qq23h2Ckyd).
>
>
> [1] Madigan, et al. "Bayesian graphical models for discrete data.", International Statistical Review/Revue Internationale de Statistique,1995.
>
> [2] Su, et al. "Improving structure mcmc for bayesian networks through markov blanket resampling.", The Journal of Machine Learning Research, 2016.
>
> [3] Kuipers, et al. "Partition MCMC for inference on acyclic digraphs.", Journal of the American Statistical Association, 2017.
>
> ---
>
> > **Q2: What is a good criterion (or number of samples) to ensure proper convergence for the constructed Markov chain(s)?**
>
> - We assume you are inquiring about methods for diagnosing Markov chain convergence.
> - Convergence diagnosis is a complex issue, and multiple tests such as _Heidelberger_ and _Geweke_ can be used to validate convergence from different angles, though they come with additional computational and time costs.
> - A common approach is to increase the number of steps to pass through the burn-in phase, but it is important to note that the convergence speed of MCMC-generated chains is highly sensitive to the initial state. Therefore, we cleverly select a good initial state through a proxy algorithm to ensure faster convergence.
> - For a more detailed discussion, please also refer to our  [_**"General Response to All Reviewers Regarding the Convergence of Our IS-MCMC Algorithm"**_](https://openreview.net/forum?id=ZXs3pkmrRG&noteId=xecRfSr4VW).
>
> ---
>
> > **Q3: Why is the use of a binary classifier for CI tests preferable? Are there empirical evidences supporting this?**
>
> - Here, we provide a simple example by randomly selecting 30 re-generated $D_t$ datasets and running statistical averages and means using the standard PC algorithm, which we term "Proxy-PC". The results, compared with "JCI-PC" and our method on the $\mathcal{I}$-CPDAG task, are as follows:
>
> |  |  | Survey |  |
> | --- | --- | --- | --- |
> |  | SHD | SID | F1 |
> | Proxy-PC | 7.17±1.70 | 19.51±7.23 | 0.33±0.13 |
> | JCI-PC | 5 | 18 | 0.40 |
> | Our | 0 | 0 | 1.0 |
>
> - In **Appendix D** of the revised version, we provide a systematic study from intuitive analysis to theoretical proof of the superiority of our method over the PC algorithm. We also recommend reviewing our [_**"General Response to All Reviewers Regarding the Identifiability of Our PC-like SCL Algorithm"**_](https://openreview.net/forum?id=ZXs3pkmrRG&noteId=JzO7AtIHvC).
> - In brief, our _**PC-Like SCL**_ algorithm, while maintaining theoretical asymptotic correctness, encourages a more systematic characterization and better classification mechanisms, making it more robust than the PC algorithm, and thus yielding superior empirical performance.

---

> ### Author Response · Authors · 2024-11-25
> **The Second Part of the Response to Reviewer 4MTP**
>
> ---
>
> > **Q4: Why is the sample size of 10,000 chosen for forward sampling? Is there a risk of overfitting? Why does performance improve with larger sample sizes?**
>
> - We follow the default settings from previous studies to select the sample size for ancestral sampling (please refer to **Table 6** in **Appendix C.2**, which summarizes the configurations of all different methods).
> - Additionally, in **Table 9** of **Appendix C.4**, we systematically summarize the configurations of our extensive experiments, including the effects of varying observation and intervention sample sizes.
> - In direct response, the goal of our paradigm is to generate as much data as possible for model fitting, thereby avoiding out-of-distribution generalization issues. Hence, larger sample sizes typically lead to better performance, as evidenced by the comparisons in **Tables 1** and **16**.
>
> ---
>
> **We deeply value your feedback** and will incorporate these detailed discussions into the final revision of the paper. We hope the above responses address your concerns. _**If possible, we kindly request that you reconsider the score.** Should you have any further suggestions, we would be happy to discuss them and make any necessary improvements to the paper._
>
> Best wishes
>
> Authors of submission 3081

---

> > ### Comment · Reviewer_4MTP · 2024-11-26
> > **Re.**
> >
> > Thanks to the authors for their reply. Most of my concerns are addressed by these comments. Incorporating these detailed discussions into the final revision of the paper will improve it significantly. I will increase my score to a 6.

---

> ### Author Response · Authors · 2024-11-26
> **Sincere Gratitude from Authors**
>
> Dear Reviewer 4MTP,
>
> **We sincerely appreciate the time you spent** reviewing our responses and your willingness to raise your score in recognition of our work.
>
> We are pleased to hear that our rebuttal has effectively addressed most of your concerns, given the complexity of the material we need readers to understand.
>
> ---
>
> We would like to summarize **the revisions made for you during this period** (please refer to the [_**```current revised version```**_](https://openreview.net/pdf?id=ZXs3pkmrRG) with changes highlighted in purple):
>
> + We have rewritten the _**IS-MCMC process**_ described in **Algorithm 1** and revised the related **Figure 3** to include corresponding labels, making it easier for readers to follow the process. Furthermore, we have added **new Appendix C**, which specifically discusses the convergence of IS-MCMC and the motivations and considerations behind convergence optimization. _Additionally, we plan to explore further improvements and experiments in this algorithm in the future_.
>
> + In response to your concerns regarding the _**advantages of the modified PC algorithm**_ and, most notably, its identifiability, we have added a **new Appendix D**. This section provides a detailed analysis of the _**observations, motivations, and theoretical proofs**_ to address the concerns related to the PC-like algorithm.
>
> ---
>
> We also **warmly invite you to review our general response**, which offers a quick and clear overview of the positioning and contributions of our work. We believe our progress is worth sharing with the community. _Should you have any further suggestions, we would be happy to hear them and continue improving our work in next days._
>
> Once again, Thank you!
>
> Sincerely,
>
> Authors of submission 3081

---

### Official Review · Reviewer_ya31 · 2024-11-06

**Soundness:** 3
**Presentation:** 1
**Contribution:** 2
**Rating:** 3
**Confidence:** 3

**Summary:**

This paper proposes an algorithm that performs causal discovery from observational and interventional data while adapting to test-time training. It is built upon the joint causal inference and supervised causal learning frameworks. The authors provide their performance results on a variety of causal graphs.

**Strengths:**

The authors address a very interesting and important problem. The experiments are extensive, and the necessary background to understand the paper is provided.

**Weaknesses:**

## Minor Weaknesses:
* The introduction discusses many contributions, which might be a little distracting.
* Figure 2's caption should mention what the numbers in the curly brackets refer to.
* $X_2 \to X_1 \leftarrow \\{1\\}$. It is unclear why $\\{1\\}$ is considered a node here.
* Assumptions such as Markovian, faithfulness, sufficiency, etc. should be defined and explained.
* I-CPDAG is not discussed in detail, although it is used many times.
* JCI considers data samples under interventions as observations under imposed conditions — this statement appears repetitive.
* $D_i$ is likely to be in the neighborhood of the original augmented dataset. It is unclear what "neighborhood" means.
* It is unclear what "high-quality training data" means. This should be made precise.
* It is not properly defined what $G^{(t-1)}$ refers to in the mutation step.
* The numbers 1-6 in the key steps could be shown in Figure 3 for better mapping.

## Major Weaknesses:
* It is hard to focus on the main contribution: Is it I-CPDAG discovery, finding the intervention target, test-time adaptation, PC+SCL, prior knowledge, etc.?
* The presentation of the paper needs to improve a little more. It feels like too many things are going on. The description needs to be clearer.
* Although test-time adaptation was the main motivation of the proposed approach, it is discussed very little.
* Are the authors proposing to train a classifier for each possible triplet to detect if it forms a v-structure? That would be very computationally inefficient. How is that better than using the PC algorithm? The training procedure needs more detailed discussion.
* The Asia graph from bnlearn is not a real-data graph. Please check its reference.

I highly appreciate the author's effort on this paper. I hope my comments help them to improve it further.

**Questions:**

## Questions:
* If the $\text{score}(G^{\text{cand}}) < \text{score}(G^{(t-1)})$ and $\frac{\text{score}(G^{\text{cand}})}{\text{score}(G^{(t-1)})} = 0.9$, does that mean $G^{\text{cand}}$ is chosen with probability 0.9 even if its score is lower? Doesn't that mean we will gradually move toward incorrect graphs?
* To my knowledge, the 14 graphs from bnlearn are mainly observational (except Sachs). How are the authors generating observational and interventional datasets from these graphs?

---

> ### Author Response · Authors · 2024-11-25
> **The First Part of the Response to Reviewer ya31**
>
> Dear Reviewer ya31,
>
> We sincerely appreciate the time and effort you invested in providing insightful feedback on our paper. We are also glad to see that you recognize several key aspects of our work, such as the _**"very interesting and important problem"**_,  _**"extensive experiments"**_, and _**"thorough background"**_. We have carefully considered each of your detailed suggestions and addressed them one by one to alleviate any concerns. Specifically:
>
> ---
>
> > **Major-W1: The main contributions are unclear: Is it I-CPDAG discovery, intervention target identification, testing time adaptation, PC+SCL, or prior knowledge, etc.?**
> >
> - We apologize for any confusion caused and appreciate your recognition of the importance of this problem.
> - Based on feedback from all reviewers, we acknowledge that we have covered too many aspects, which may have made it difficult for readers to identify our core contribution.
> - To clarify our work and make our contributions more discernible, we have rewritten the introduction section to systematically outline our work from the perspectives of _**problem**_, _**idea**_, and _**technique**_. This should help the reader better understand our paper. Please refer to our [_**"General Response to All Reviewers Regarding the Main Contributions of Our Paper"**_](https://openreview.net/forum?id=ZXs3pkmrRG&noteId=Qq23h2Ckyd).
> - In short, our core contribution lies in utilizing _**Supervised Causal Learning (SCL)** tools_ to address _**Interventional Causal Discovery (ICD)** problem_, for which we propose the novel _**Test-Time Training (TTT) + Joint Causal Inference (JCI)** paradigm_ and explore _customized **IS-MCMC**_ and _**PC-Like SCL** algorithms_ for training data acquisition and model learning.
>
> ---
>
> > **Major-W2: The paper’s presentation could be improved; too many things are happening.**
> >
> - We believe the changes made in response to **Major-W1** effectively address this concern. Specifically, we have revised the introduction to emphasize and highlight the key contributions of our method.
>
> ---
>
> > **Major-W3: Although testing time adaptation is a key motivation, it is discussed minimally.**
> >
> - We believe the revisions made in response to **Major-W1** also address this concern. We have clarified in the introduction why testing time training is necessary.
>
> ---
>
> > **Major-W4: Are the authors proposing to train a classifier for each possible triplet to detect whether it forms a v-structure? This would be highly inefficient. How does it compare to the PC algorithm? The training process requires more detailed discussion.**
> >
> - We do not train a classifier for each possible triplet to detect v-structures. Instead, we train a classifier on triplet features extracted from all training data to detect whether any new triplet forms a v-structure.
> - Our method is not inefficient. As shown in the revised version **Figure 8 (b)**, our method demonstrates a runtime similar to that of non-learning methods while achieving better performance.
> - In addition, we provide a detailed theoretical analysis of the superiority of our approach over the PC algorithm in **Appendix D** of the revised version, and we encourage you to refer to our [_**"General Response to All Reviewers Regarding the Identifiability of Our PC-like SCL Algorithm"**_](https://openreview.net/forum?id=ZXs3pkmrRG&noteId=JzO7AtIHvC).
> - In short, our PC-like SCL algorithm, while preserving theoretical correctness, encourages a more systematic feature representation and learns better classification mechanisms, making our method more robust than PC and thus leading to superior empirical performance.
>
> ---
>
> > **Major-W5: The Asian network in bnlearn is not a real-world data graph.**
> >
> - We apologize for any confusion caused by our wording.
> - The bnlearn repository contains various causal graphs inspired by real-world applications and has been referred to as a "real benchmark" in previous literature [1, 2, 3]. While these also can be seen as semi-real synthetic datasets generated from real causal graphs.
> - Following your suggestion, we have thoroughly reviewed and revised the wording throughout the paper regarding "real-world datasets".
>
> [1] Ke et al., Neural Causal Structure Discovery from Interventions, TMLR, 2023.
>
> [2] Ke et al., Learning to Induce Causal Structure, ICLR, 2023.
>
> [3] Lippe et al., Efficient Neural Causal Discovery without Acyclicity Constraints, ICLR, 2022.

---

> ### Author Response · Authors · 2024-11-25
> **The Second Part of the Response to Reviewer ya31**
>
> ---
>
> > **Minor-W1: The contributions discussed in the introduction are too many and somewhat distracting.**
> >
> - We believe our response to **Major-W1** addresses this concern effectively. We have reworked the introduction to focus more clearly on our key contributions.
>
> ---
>
> > **Minor-W2: What do the numbers in the curly braces of Figure 2 represent? They should be explained.**
> >
> - The numbers in the curly braces of **Figure 2** represent intervention variables (or environmental nodes). We apologize for any confusion, as we may have assumed a higher level of background knowledge of Joint Causal Inference from the reader. **Reviewer dQ5H** also pointed out this weakness.
> - We appreciate your suggestion and have moved this explanation to the last section of the **Preliminaries**, naming it **"Identifiability with Intervention Data"**, to improve clarity and reader experience.
>
> ---
>
> > **Minor-W3: It is unclear why {1} is considered a node here.**
> >
> - We believe our response to **Minor-W2** clarifies this confusion. More directly, this is default setting in the context of JCI.
>
> ---
>
> > **Minor-W4: Definitions of Markov, faithfulness, and sufficiency should be provided for clarity.**
> >
> - These concepts have been clearly defined in the **Preliminaries** section and **Appendix A**:
>   - A joint probability distribution $P_{\mathbf{X}} $is **Markov** compatible with  $\mathcal{G} $, i.e.,  $P_{\mathbf{X}} = \prod_{i=1}^{d} P(X_i | pa(X_i)) $, where  $pa(X_i) $ are the parents of  $X_i $.
>   - **Faithfulness** assumption:  $X \perp_P Y | Z \Rightarrow X \perp_{\mathcal{G}} Y | Z$.
>   - **Causality sufficiency**: Exogenous variables are ignored.
>
> ---
>
> > **Minor-W5: The I-CPDAG is not discussed in sufficient detail.**
> >
> - We have clearly described the $\mathcal{I}$-CPDAG as an instance of the I-MEC in the problem summary.
> - Our main text: _"We aim to predict all causal relations entailed by the given data $\mathcal{D}$ (subject to the above assumptions), which correspond to the invariant causal structures in the I-MEC set of the causal graph $\mathcal{G}$ behind $\mathcal{D}$, as explained in Section 2. Such invariant causal structures can be computationally encapsulated as a partial DAG, called the Interventional-Complete Partial Directed Acyclic Graph ($\mathcal{I}$-CPDAG), in which each directed edge indicates an invariant causal relation in the I-MEC set."_
>
> ---
>
> > **Minor-W6: The phrase 'JCI considers data samples under interventions as observations under imposed conditions' seems redundant.**
> >
> - Thank you for your feedback. We have rewritten the phrasing for clarity.
>
>
> ---
>
> > **Minor-W7: The meaning of "high-quality training data" is not sufficiently clear. This should be clarified.**
> >
> - This refers to the training data mentioned in the "What" stage of Test-Time Training.
> - Thank you for your reminder; we have revised the wording accordingly to ensure clarity.
>
> ---
>
> > **Minor-W8: $\mathcal{G}_i$ is likely in the "neighborhood" of the original augmented dataset. It is unclear what "neighborhood" means.**
> >
> - We apologize for the confusion. We meant that $\mathcal{G}_{i}$ is likely to maintain a certain "similarity"  with the original augmented graph $\mathcal{G} _ {\mathcal{I}}$, which entails the similarity properties of the data $\mathcal{D}_i$.
> - We have rewritten this for better clarity.
>
> ---
>
> > **Minor-W9: The meaning of $\mathcal{G} ^ {(t-1)}$ in the mutation step is not properly defined.**
> >
> - We apologize for the confusion in the algorithm workflow description.
> - Based on your feedback, we have rewritten **Algorithm 1** to clearly map the notation to the revised steps, ensuring it is easier to understand.
>
> ---
>
> > **Minor-W10: The numbers 1-6 in the key steps could be shown in Figure 3 for better mapping.**
> >
> - Following your suggestion, we have added the numbers 1-6 to **Figure 3** in the revised version for clearer mapping to the steps.

---

> ### Author Response · Authors · 2024-11-25
> **The Third Part of the Response to Reviewer ya31**
>
> ---
>
> > **Q1: Gradually converging to incorrect graphs?**
> >
> - Our approach follows the standardized Structural-MCMC process, converging to the posterior distribution $P(G | D)$. Thus, from a theoretical perspective, we do not converge to incorrect graphs.
>   - Note 1: With standard MCMC-based training data, we demonstrate consistent performance, as shown in **Figure 5**.
>   - Note 2: While MCMC guarantees convergence, some local fluctuations may occur, which relate to optimizing the MCMC algorithm—a topic for future work.
> - For a detailed discussion on MCMC convergence, please refer to our  [_**"General Response to All Reviewers Regarding the Convergence of Our IS-MCMC Algorithm"**_](https://openreview.net/forum?id=ZXs3pkmrRG&noteId=xecRfSr4VW).
>
> ---
>
> > **Q2: The 14 graphs in bnlearn are mainly observational (except for Sachs). How do the authors generate observational and intervention data from these graphs?**
> >
> - The bnlearn repository includes causal graphs inspired by real-world applications, and these can be considered semi-realistic simulator data based on true causal graphs.
> - Each causal graph in bnlearn provides conditional probability distributions (CPDs), allowing us to easily simulate intervention experiments. Specifically, we implement both hard and soft interventions.
>   - A hard intervention directly sets the value of a variable, ignoring its normal distribution. Our code implementation for hard intervention is: `hard_intervention[var] = self.bn.states[var][np.random.randint(0, self.bn.get_cardinality(var))]`.
>   - A soft intervention adjusts the CPD of a variable to influence its distribution. We implement this via a Dirichlet distribution for the target variable: `values = [[x] for x in np.round(np.random.dirichlet(np.ones(cardinality) * probability), 2)]`.
>
> ---
>
> Once again, **we greatly appreciate your valuable feedback** and have incorporated these discussions into the final revision of the paper. We hope the above responses address your concerns. ***If possible, we kindly request that you reconsider the score. Should you have any further suggestions, we are more than willing to discuss them and make necessary improvements.***
>
> Best wishes
>
> Authors of submission 3081

---

> ### Author Response · Authors · 2024-11-27
> **Kindly Request for Reviewer's Feedback**
>
> Dear Reviewer ya31,
>
> Since the End of the Rebuttal is coming very soon - only a few days left, we would like to inquire _whether our responses have addressed your major concerns_.
>
> ---
>
> We also _**cordially invite you to review our general response**_, which quickly and clearly outlines the positioning and contributions of our work. We would like to emphasize that proposing the use of the TTT+JCI paradigm through SCL tools to address ICD problem is _**by no means trivial**_. Currently, the state-of-the-art solutions for this problem, such as CSIvA[1] and AVICI[2], do not take test-time training into account, which _**means they are unable to avoid the generalization problem**_. Moreover, they can only handle limited intervention settings and thus are _**unable to adapt to diverse intervention scenarios**_.
>
> _**We have incorporated this perspective and updated our related work in the [_**```current revised version```**_](https://openreview.net/pdf?id=ZXs3pkmrRG) in purple font.**_
>
> ---
>
> We believe that our progress is worthy of being shared with the community. If you have any further suggestions, we will be glad to listen to them and continue to improve our work in the coming days.
>
> We look forward to your feedback.
>
> Sincerely,
>
> Authors of submission 3081
>
> ---
>
> [1] Ke et al., "Learning to Induce Causal Structure", ICLR, 2023.
>
> [2] Lorch et al., "Amortized inference for causal structure learning", NeuIPS, 2022.

---

> ### Comment · Reviewer_ya31 · 2024-11-27
>
> I cordially thank the authors for their details response and appreciate their effort for this paper. However, I personally think the paper needs in-depth look and some major revision. Thus, I prefer not to increase the score.

---

> ### Author Response · Authors · 2024-11-28
> **Has our revised manuscript addressed your concern?**
>
> Thank you for taking the time to respond.
>
> Regarding ```"the paper needs in-depth look and major revision"```, we feel there might be a misunderstanding here -- in fact we have already uploaded a revised manuscript of the paper. The revised paper has articulated a concise summary of our main contributions (which seems to be your main concern in the review). So by "major revision", do you mean further revision on top of this revised manuscript? Have our existing revisions and responses addressed your main concerns? If not, we would highly appreciate it if you could outline your remaining concerns at the moment. This could help us better understand the issues and make meaningful improvements to our work.
>
> In any case, we want to make sure that you didn't miss the revised manuscript, where the introduction, related work, and analysis of IS-MCMC / PC-like SCL (Appendix C and D), has been significantly revised particularly for addressing your specific concerns.

---

### Official Review · Reviewer_dQ5H · 2024-11-10

**Soundness:** 3
**Presentation:** 4
**Contribution:** 3
**Rating:** 8
**Confidence:** 3

**Summary:**

The authors focus on the setting where we have multiple datasets describing the same system, but each dataset was collected under a different set of interventions, whose targets may be unknown.  This problem setting matches that of the joint causal inference framework proposed by Mooij et al., which provides a framework for combining such a collection of datasets and conceptualizing them with a single graph.  The authors propose a novel method, TICL, that first generates semi-synthetic data from from a graph structure learned from the unified JCI datasets, producing a training dataset.  TICL then uses this data to in a two-phase PC-like supervised causal learning process, using classifiers to determine first edge existence and then V-structure orientation.  The authors compare TICL to a variety of other methods on bnlearn baselines and find that it consistently performs the best.

**Strengths:**

I think this is a good paper.  The authors take two existing techniques (JCI and SCL), describe both of them well, and provide an interesting combination that both feels intuitive and clever.  The writing is, for the most part, clear throughout, with consistent terminology and notation.  The experimental results, while only semi-synthetic (due to the lack of real-world interventions), are thorough and compelling, comparing against a solid range of competitive methods in multiple categories.

**Weaknesses:**

I appreciate the attempt at a "When", "What", "How" framing in the introduction, but the framing of "We create training data after accessing the test data" is strange and over-complicates what you're actually doing.  If I understand correctly, your approach is a two-stage process, using the input data $D$ to generate augmented semi-synthetic data, which is used to train a model that to classify $D$ in your second stage.  I think part of the confusion is the use of the term "test data" here, since in ML, tests data is usually a dataset that is used at the very end of a pipeline to assess model performance, which doesn't really match the standard causal inference setting.  I didn't understand the text of the "When" section in the introduction at all until I had nearly finished the paper, and even then it took multiple times reading it to understand what you were trying to say.  Rewording that section to focus on how the data that is typically use for SCL training differs from what you use, and how that allows you to train an SCL model that is far more tailored to your specific problem.

Not being very familiar with the JCI framework before reading this paper, I found the organization around Sections 3.1 and 3.2 confusing.  Figure 2, and the discussion of it at the end of 3.1, is very helpful, but it makes use of the JCI framework (referencing intervention families and showing how intervention targets are present in the graph).  This figure makes a lot of sense after reading Section 3.2, but since it's presented and discussed in 3.1 (even using terms like "intervention family" before they are defined), it mostly serves to confuse and force the reader to try to understand the JCI framework from context clues.  Moving the final paragraph of 3.1 until after Section 3.2 would help a lot.

While I generally like the experimental results, some of the discussion there feels a bit disingenuous.  The authors state one of their key advantages  over other recent methods is that you tested on "real causal graph datasets with over 100 nodes", and in the "Benchmarks & Baselines" section, you say that you "use discrete datasets of 14 real-world datasets collected from the bnlearn repository".  This implies that all 14 of the datasets are "real-world datasets", many of which are over 100 nodes.  However, unless I'm missing some experiments or missing something about the bnlearn datasets, the only one of the 14 you list that has over 100 nodes is Pathfinder, which has 104, and some of the bnlearn datasets you chose (Asia, Insurance, Alarm, Hailfinder) are synthetic, so calling them "real-world datasets" is strange.
 You definitely test on a broader array of datasets than "only synthetic with 10-30 nodes", so I think that's a fine distinction to highlight, but it would be helpful, when describing the datasets you used in the "Benchmarks & Baselines" section, to be clearer about the range of nodes and realism of the datasets you chose.

I understand your reasoning, but it's a bit strange to call AVICI and ENCO "unfair baselines".  While they are told the intervention targets and other methods are not, TICL and the other JCI-based methods also have the distinct advantage that the data you are testing on is data from bnlearn that you modified (via interventions) to exactly match the problem setting for the JCI framework, upon which TICL is based, so it's not surprising that methods designed to work on data with multiple interventions with unknown targets perform better than methods intended to work on data with either a single intervention or known intervention targets.  You of course should make it clear to the reader that AVICI and ENCO had access to that information, since the fact that TICL doesn't need it to be competitive is a clear advantage, but I would drop the language about them being unfair baselines because of it.

A real-world example would help a lot towards showing the value of this method.  While the method seems interesting, the setting of "We have both observational and interventional data that measures the same variables, but the targets of each intervention are unknown" comes across as very niche and not very applicable.  I don't think that's necessarily the case, but providing even a simple motivating example in the introduction would help explain to the readers why we should care about your method.

In Table 1, the meaning of blue/bold and red/underlined should be included in the Table 1 caption, rather than in the "Training Datasets & Metrics" section, so the table can be at least somewhat stand-alone.  I would also put Footnote 1 on the table, rather than at the beginning of the list of methods (especially since you don't actually use the * until the table).

A minor point, but in Table 1, it would be good to use some other indicator for "JCI-Improved baseline", "Unfair baseline", and "classic baseline" besides a light background color.  As is, it's impossible distinguish when printed in black and white and even in color, the light yellow for "classic baseline" blends in far too much with the background and looks too similar to the pale green.  Similarly, in the two right plots of Figure 4, it's odd to use different symbols and colors for the points, but only include the colors (and not the symbols) in the legend.  You should make the legend show the shapes as well.

**Questions:**

Can you clarify the loop structure of Algorithm 1?  The parallelization loop index, $i$, is not used in the algorithm pseudo-code, and if that loop is being done in parallel, then presumably there's no looping back to step 1 from step 7, right?  But then step 1 starts with us already having $G^{t-1}$.  Where is $t$ initialized?  I wondered if maybe steps 1-7 were being done iteratively until IS-MCMC converges, but then I'm not sure what's happening with the $D^{(t)}$ that's sampled in step 7, since if we're making a list of each $D^{(T)}$ dataset for each variant of $G^{(t)}$ we see, then we'd probably want some sort of burn-in period, which isn't part of the algorithm.  But if we're just keeping the final $D^{(t)}$, then there's no reason for it to be in a loop... Clearly there's something I'm missing with how this algorithm is set-up.

You mention that you consider soft-interventions for your experiments.  Can you give an example of the type of soft intervention you simulated?  Did you try experiments with hard interventions, and, if so, were the results essentially the same?

Did you try comparing against generating $D^{(t)}$ from Algorithm 1 and then just using standard PC on it, rather than your custom PC-like approach?

---

> ### Author Response · Authors · 2024-11-25
> **The First Part of the Response to Reviewer dQ5H**
>
> Dear Reviewer dQ5H,
>
> We sincerely appreciate the time and effort you have spent providing insightful feedback on our paper. We are also pleased that you recognize several strengths of our work, including its _**"interesting, intuitive, and clever idea"**_, _**"clear writing"**_, and _**"thorough and compelling experiments"**_. We have carefully considered each of your detailed comments and have addressed them one by one to alleviate your concerns. Specifically:
>
> ---
>
> > **W1:  The "when," "what," and "how" sections are well written, but the phrasing of the "when" part is confusing, as it conflates the term "test data". The expression needs to be revised for clearer understanding.**
>
> - We greatly appreciate your recognition of our writing style and apologize for any confusion caused.
> - Based on your suggestion, we have revised the wording here. Specifically, we define two stages during the "test time": "accessing the test data" and "performing the actual test." Our method operates between these two stages—after accessing the test data but before performing the actual test—by generating free training data, training the model, and using it for final testing.
> - However, based on feedback from other reviewers, we realized that many readers might struggle to follow the core ideas of the paper. Therefore, _we have rewritten the Introduction to systematically clarify our work from the perspectives of the **problem**, **idea**, and **technique**_, which we believe will aid the readers' understanding. Please see our [_**"General Response to All Reviewers Regarding the Main Contributions of Our Paper"**_](https://openreview.net/forum?id=ZXs3pkmrRG&noteId=Qq23h2Ckyd).
>
> ---
>
> > **W2: The organization of Sections 3.1 and 3.2 in original draft is confusing, and moving the last paragraph of 3.1 to the end of 3.2 would be very helpful.**
>
> - Thank you for your feedback. It is important to note that we defined and used terms such as "intervention family" in the second-to-last paragraph of Section 3.1 in the previous version, where we aimed to describe the identifiability using intervention data.
> - However, we agree with your suggestion. We have moved this content to the end of the "Preliminaries" section and renamed it **"Identifiability with Intervention Data"** to enhance clarity and improve the reading experience.
>
> ---
>
> > **W3: The wording seems misleading, as the only dataset with more than 100 nodes, Pathfinder, and some of the datasets in bnlearn are synthetic.**
>
> - We apologize for the misleading wording. Our intention was to select causal graphs of various sizes and challenges to emphasize the difficulty of causal discovery under our setting, which prior works rarely valued.
> - The bnlearn repository contains causal graphs inspired by real-world applications, often referred to as "real-world benchmarks" in previous literature [1, 2, 3]. While these also can be seen as semi-real synthetic datasets generated from real causal graphs.
> - Based on your suggestions, we have thoroughly checked all inappropriate wordings regarding the dataset description and revised them in the new version.
>
> [1] Ke et al., "Neural Causal Structure Discovery from Interventions," TMLR, 2023.
>
> [2] Ke et al., "Learning to Induce Causal Structure," ICLR, 2023.
>
> [3] Lippe et al., "Efficient Neural Causal Discovery without Acyclicity Constraints," ICLR, 2022.
>
> ---
>
> > **W4: Referring to AVICI and ENCO as "unfair baselines" is somewhat strange, and the phrase should be abandoned.**
>
> - Thank you for your suggestion. We have adopted your recommendation and no longer refer to AVICI and ENCO as "unfair baselines." Instead, we have renamed them as "Modified adaptation baseline" in the revised manuscript.
>
> ---
>
> > **W5: Providing a real-world example would be helpful. The paper presents both observational and intervention data for the same variables, but the targets of the interventions are all unknown, which may seem a niche setting.**
>
> - It is important to note that our setting is not particularly niche. On the contrary, it is one of the most comprehensive and rich settings we are aware of, as demonstrated in **Appendix E.2, Table 6** and **Appendix E.4, Table 9**.
> - Additionally, you may have missed further experimental results in **Appendix F**, which include varying types of intervention targets (hard vs. soft, single vs. multiple), different sample sizes for observational and intervention data, and the number of intervention experiments.
> - If you were referring to experiments with known intervention targets, this can be viewed as a simpler special case of our method. As described in the last paragraph of **Section 3.2**, when the intervention targets are known, the prior knowledge from the augmented graph can directly identify the environmental nodes pointing to the system nodes of the intervention target.

---

> > ### Comment · Reviewer_dQ5H · 2024-11-26
> >
> > I appreciate the authors' response and willingness to change the paper to improve clarity.  The inclusion of % bias is a nice addition to the results, but I still wish the authors had come up with some baseline to assess performance.  Given that these are simulation results, it's hard to know how much of the low % bias is because the algorithm is amazing and how much is because the problem is easy.  Oddly, the table of experimental results (Table 1) is now incorrectly referred to in the text as Table 3.  It looks like it's correct in the original submitted version, so I'm not sure why that would have changed the table number...
> >
> > The paper has been improved some with the edits, so I am willing to improve my score from a 3 to a 5.  Ultimately, however, even with the changes, the paper suffers from clarity issues.  For example, the new motivating example in the intro is a better example, but now the paper starts with just "As a motivating example", rather than actually describing the problem, making it confusing now for a different reason.  I do think there is good content here, and if the other reviewers believe the clarity issues aren't as big a deal, I'm not going to oppose accepting this paper.  As it stands, though, I think this paper needs multiple additional editing rounds.

---

> ### Author Response · Authors · 2024-11-25
> **The Second Part of the Response to Reviewer dQ5H**
>
> ---
>
> > **W6: The meanings of blue/bold and red/underlined text, as well as **Footnote 1**, should be included in the table's caption so that the table can stand alone.**
>
> - Thank you for your suggestion. We have included the meanings of blue bold and red underlined text in the caption of **Table 1**. However, regarding **Footnote 1**, since it does not relate to the information in the table and the only related baseline adjustment is already changed in **W4**, we have decided to leave it unchanged.
>
> ---
>
> > **W7: The colors of different categories of methods are difficult to distinguish, and ensure that the legend and icons in Figure 4 align correctly.**
>
> - Thank you for your suggestion. We have implemented your recommendation by enhancing the background color distinctions in **Table 1** and ensuring the legend and icons in **Figure 4** are aligned.
>
> ---
>
> > **Q1: Clarify the loop structure in Algorithm 1 and explain how it is parallelized.**
>
> - We apologize for the confusion caused by the algorithm flow, which mixed practical aspects. In reality, your understanding is correct: Steps 1–6 in original manuscript iterate continuously, while Step 7 is sampled after the burn-in period, and these two components operate in parallel processors. The index $i$ is not required in the loop, indicating internal parallelism.
> - To clarify, we have rewritten **Algorithm 1** in the revised manuscript and aligned it with the notation introduced in the main text. We believe the revised algorithm flow is now sufficiently clear and easy to understand.
>
>
> ---
>
> > **Q2: Could you provide an example of the soft interventions you simulate? Have you tried hard intervention experiments? If so, are the results similar?**
>
> - **Hard intervention** refers to directly setting the value of a variable. In other words, a hard intervention forces the target variable to assume a specific value, disregarding its normal probability distribution. Specifically, our code implementation is: `hard_intervention[var] = self.bn.states[var][np.random.randint(0, self.bn.get_cardinality(var))]`, which randomly selects a state value for each target variable and assigns it as the value for the hard intervention.
>
> - **Soft intervention** is a more subtle form of intervention. Instead of directly setting a variable’s value, it influences its distribution by adjusting its Conditional Probability Distribution (CPD). Specifically, our code implementation is: `values = [[x] for x in np.round(np.random.dirichlet(np.ones(cardinality) * probability), 2)]`. Here, we use the Dirichlet distribution to generate a new probability distribution for the target variable. The Dirichlet distribution is suitable for discrete data and multiclass problems, as it effectively generates a set of non-negative values that sum to 1, representing a valid probability distribution.
>
> - For the _**hard intervention experiments**_ , detailed results and analysis can be found in _**Appendix F.1.1**_, _**Table 14**_ of the revised version. Comparing this with _**Table 1**_, the results are similar.
>
> ---
>
> > **Q3: Have you tried comparing the results from Algorithm 1 generating $D_t$ and then using the standard PC instead of your custom PC-like SCL method?**
>
> - Yes, this typically yields better performance. We provide a simple example where we randomly select 30 generated  $D_t$ samples and run standard PC, reporting the statistical averages as "Proxy-PC," which is compared with JCI-PC and our method on the $\mathcal{I}$-CPDAG task performance, as follows:
>
> | Method      | SHD   | SID   | F1   |
> |-------------|-------|-------|------|
> | Proxy-PC    | 7.17±1.70 | 19.51±7.23 | 0.33±0.13 |
> | JCI-PC      | 5     | 18    | 0.40 |
> | Ours        | 0     | 0     | 1.0  |
>
> - In **Appendix D**, we provide a systematic study from intuition to theoretical proofs to demonstrate the superiority of our method over the PC algorithm. Please also refer to our [_**"General Response to All Reviewers Regarding the Identifiability of Our PC-like SCL Algorithm"**_](https://openreview.net/forum?id=ZXs3pkmrRG&noteId=JzO7AtIHvC).
>
> - In summary, _**our PC-Like SCL algorithm retains the theoretical asymptotic correctness of PC** while promoting richer feature representations and improved classifier learning, making our method **more robust and yielding superior empirical performance**_.
>
> ---
>
> Once again, _**we sincerely appreciate your valuable feedback, which has significantly strengthened our manuscript!** We hope the above responses address your concerns, and we are happy to discuss further suggestions to improve the paper._
>
> Best wishes
>
> Authors of submission 3081

---

> ### Author Response · Authors · 2024-11-26
> **A mistake in comments pasting?**
>
> Dear Reviewer dQ5H,
>
> We have just noticed that your current comments appear to be unrelated to our paper. Could you kindly confirm if you might have accidentally responded regarding another paper you are reviewing?

---

### Author Response · Authors · 2024-11-25
**General Response to All Reviewers Regarding the Convergence of Our IS-MCMC Algorithm**

Dear All Reviewers (and AC),

We are delighted to see that most reviewers are interested in this problem. Below, we provide a point-by-point response to address the concerns raised by the reviewers:

---

> **Guarantee of Posterior Distribution Convergence:**
>
- Our IS-MCMC algorithm follows a standardized Structure-MCMC process in the intervention-augmented graph space. Please refer to **Algorithm 1** in the revised manuscript, which corresponds to **Steps 1-6** in the main text as a general introduction to the Structure-MCMC process.
- By employing the Metropolis-Hastings sampler within the standardized Structure MCMC framework, the Markov chain is guaranteed to have a stationary distribution equal to the posterior distribution $P(G \mid D)$ [2], which has been well studied in the literature[1, 2, 3].
- It is important to note that, under the premise of this theoretical guarantee, the novelty of our work lies in proposing the use of the posterior distribution as the target for acquiring training samples and optimizing feasible solutions in intervention scenarios.


[1] Madigan, et al. "Bayesian graphical models for discrete data.", International Statistical Review/Revue Internationale de Statistique,1995.

[2] Su, et al. "Improving structure mcmc for bayesian networks through markov blanket resampling.", The Journal of Machine Learning Research, 2016.

[3] Kuipers, et al. "Partition MCMC for inference on acyclic digraphs.", Journal of the American Statistical Association, 2017.

---

> **Optimization Considerations for IS-MCMC:**
>
- However, two critical factors need to be addressed.
  - **First**, managing the time complexity of this process is crucial for maintaining efficient inference.
  - **Second**, ensuring the effectiveness of the IS-MCMC process on the augmented graph.
- We try to solve the above two problems:
  - **From the TTT-driven perspective on efficiency:** Since we leverage the "test-time" phase, the convergence speed of IS-MCMC is particularly important. We observed that a good initial state enables the Markov chain to converge faster, as evidenced by **Figure 5** in the experimental section of our paper. Additionally, we modified the algorithm to use parallel multi-chain sampling, allowing for further efficiency, as shown in **Figure 8** of our experimental results.
  - **From the JCI-driven perspective on adaptability:** The MCMC process operates on augmented graphs rather than standard causal graphs. Thus, additional constraints are required to ensure the validity of the augmented graph. To address this, we introduced intervention constraints to ensure consistency of the augmented graph within the JCI framework.

---

In summary, our IS-MCMC algorithm effectively addresses multiple challenges under the TTT + JCI framework and demonstrates that using posterior estimates as training data is highly beneficial for test-time SCL.

As a practical response, we have included a **new Section C** in the revised appendix, which further organizes these insights in detail to aid reader understanding.

Sincerely,

Authors of submission 3081

---

### Author Response · Authors · 2024-11-25
**General Response to All Reviewers Regarding the Identifiability of Our PC-like SCL Algorithm**

Dear All Reviewers (and AC),

We are pleased to see that all reviewers are interested in this issue. The main motivation for modifying the PC algorithm to the PC-Like SCL method is to leverage its powerful **error redundancy** and **theoretical identifiability**:

---

- **Empirical Performance:** In finite sample scenarios, the reliance of the PC algorithm on conditional independence (CI) tests can lead to errors. The PC-Like SCL algorithm combines feature-rich representations with a robust classification mechanism, achieving outstanding empirical results when compared to JCI-PC or other non-SCL methods. This is demonstrated in **Tables 1-2, 10-17** and **Figure 13** of our paper.

- **Theoretical Guarantees:** Our PC-Like SCL method **retains** the asymptotic properties of the original PC algorithm. Specifically, when the sample size approaches infinity, the method will detect the correct $\mathcal{I}$-CPDAG in the intervention environment, ensuring theoretical identifiability.

---

More specifically, our PC-Like SCL method encompasses the following ideas:

- **Feature Extraction:**
    - PC-Like SCL algorithm extracts a richer set of features that capture the conditional dependencies and structural patterns around training instances.

- **Classifier:**
    - Unlike heuristic rules in methods like PC or MPC, we train a classifier on synthetic data. Our model learns the complex interactions between features, overcoming the limitations of CI tests, which are prone to errors.
    - As the sample size approaches infinity, the learned classifier converges to an equivalent and theoretically correct solution, as would be provided by PC or MPC. However, in practical settings, PC-Like SCL's broader feature set and data-driven optimization enable superior empirical performance.

---

In summary, our **PC-Like SCL algorithm preserves the theoretical guarantees of PC while addressing its empirical limitations**. By combining feature-rich representations with a learned classification mechanism, PC-Like SCL achieves significant performance improvements in finite sample scenarios.

---

As a practical response, we have included a **new Section D** in the revised appendix, which explains the underlying reasons from intuitive analysis to theoretical proof to assist readers in understanding.

Sincerely,

Authors of submission 3081

---

### Author Response · Authors · 2024-11-25
**General Response to All Reviewers Regarding the Main Contributions of Our Paper——(Part 2)**

---

> **Technical Level**

While the conceptual framework unifies the paradigm, its practical implementation requires addressing specific research challenges. To this end, we design two **customized algorithms**, **IS-MCMC** and **PC-Liked SCL**, to support data acquisition and model training within the TTT+JCI paradigm.

- **Training Data Acquisition under TTT**
    - We systematically address three critical questions:
        + **Q1**. *When should training data be acquired?*
        + **Q2**. *What types of training data are effective for SCL?*
        + **Q3**. *How can such data be obtained in our context?*
    - We give the answer:
        + **A1**. We note that the "test time" can be divided into two subphases: "accessing the test data" and "performing the actual test". Our method operates between these two stages—after accessing the test data but before performing the actual test—by generating free training data, training the model, and using it for final testing.
        + **A2**. We observe that posterior estimates of causal graphs, provide highly effective training data, as evidenced by empirical results in **Section 5.2**. *(We have also added a discussion on convergence in **Appendix C** of the revised version.)*
        + **A3**. We introduce the _**IS-MCMC**_ algorithm, which constructs Markov chains in an augmented graph structure space under interventional constraints. This method efficiently samples from the posterior distribution and incorporates optimization strategies tailored for intervention settings.

- **Learning Objective Design under JCI**
    - Although the JCI paradigm simplifies the unification of diverse intervention settings, it does not directly address:
        + **Q1**. *What is the appropriate learning objective?*
        + **Q2**. *How should the learning process be designed?*
    - We give the answer:
        + **A1**. A straightforward idea is to use the generated training data to train models that directly predict causal graphs. However, we highlight the overlooked connection between identifiability in ICD and SCL: models should aim to predict identifiable causal structures, specifically **$\mathcal{I}$-CPDAG**. Recall that a causal edge is identifiable only if it consistently appears across all causal graphs compatible with the given data. Unidentifiable edges lack stable associations with the data, even if they exist in the true causal graph. *(We have also added a theoretical proof of identifiability in **Appendix D** of the revised version.)*
        + **A2**. We chose to focus on identifying identifiable components in **$\mathcal{I}$-CPDAG**, namely the **skeleton** and **v-structure**. Inspired by the PC algorithm, we adapt it into a **PC-Liked SCL algorithm**. This approach structures learning as a two-phase process: skeleton identification in the first stage, followed by orientation prediction in the second stage. This design ensures theoretical asymptotic correctness while fostering systematic feature extraction and improved classification mechanisms. Consequently, our method demonstrates greater robustness compared to PC, achieving superior empirical performance.

---

> **Conclusion**
>
In summary, our core contribution lies in utilizing _**Supervised Causal Learning (SCL)** tools_ to _address **Interventional Causal Discovery (ICD)** problem_, for which we propose the _novel **Test-Time Training (TTT) + Joint Causal Inference (JCI)** paradigm_ and explore _customized **IS-MCMC** and **PC-Like SCL** algorithms_ for training data acquisition and model learning.

---

In light of the reviewers’ feedback, we recognize that many readers may struggle to grasp the core ideas of this paper. As a concrete response, we have **rewritten the introduction section**, systematically organizing our work across the dimensions of **problem**, **idea**, and **technique** to aid comprehension.

Sincerely,

Authors of submission 3081

---

### Author Response · Authors · 2024-11-25
**General Response to All Reviewers Regarding the Main Contributions of Our Paper——(Part 1)**

Dear All Reviewers (and AC),

We extend our sincere gratitude to all reviewers for recognizing the _**significance of this research problem**_. It is crucial to emphasize that this work is not a mere incremental contribution but the outcome of a long-term and rigorous research effort, given the inherent challenges of the problem. To clarify our objectives and highlight the core contributions, we have structured our responses around three key dimensions. Specifically:

---

> **Problem Level**
>
Our aim is to address the _problem of **Interventional Causal Discovery (ICD)**_ using a _**Supervised Causal Learning (SCL)** tool_. This paper **deepens the understanding** of why SCL is used to address ICD from the _problem perspective_, and it outlines the specific challenges faced in this context.

- **Why address this fundamental problem?**
    - Causal discovery (CD) is a cornerstone of scientific inquiry. We posit that SCL is a novel and promising tool for tackling CD due to its scalability (via free training samples) and robust empirical performance. However, causal discovery using observational data is merely a special case of ICD, often hindered by limited identifiability. Addressing ICD is thus essential, yet the community has only made limited progress in realizing the potential of SCL for ICD. Existing studies, such as AVICI and ENCO, primarily focus on specific scenarios like hard interventions with known targets, leaving much of the problem space unexplored.
- **What fundamental challenges must be addressed?**
    - From the perspectives of both SCL and ICD, we identify two intrinsic challenges:
        + The *out-of-distribution generalization* issues inherent to SCL.
        + The *complexity and diversity* of ICD problems arising from different intervention experiments.

---

> **Idea Level**

To address the non-trivial challenge of leveraging SCL for ICD, we propose a **novel paradigm** combining **Test-Time Training (TTT)** and **Joint Causal Inference (JCI)**.

- **Test-Time Training for Generalization**
    - Although causal graph simulators can be designed to encompass diverse types, mechanisms, and scales, causal structure learning is inherently high-risk, often prioritizing performance at the expense of universality. SCL, due to the limitations of machine learning, inevitably faces out-of-distribution generalization issues. Instead of **_"scaling up the pre-training stage"_**, we prioritize _**"scaling up the inference stage"**_.
    - *To this end, we introduce a novel TTT framework tailored for causal discovery. By employing context-customized supervised learning during inference, this approach effectively circumvents the generalization challenges of SCL.*
- **Joint Causal Inference for Diversity**
    - Developing a unified and effective learning framework for ICD is challenging due to the diversity of intervention experiments. This diversity complicates the application of learning-based methods.
    - *We innovatively incorporate the JCI framework from traditional causal discovery, facilitating a unified learning process. This integration enables SCL tools to accommodate various intervention conditions, effectively unifying ICD. Additionally, our approach simultaneously unifies SCL for intervention target identification, which previously required bespoke algorithmic designs.*

---

### Author Response · Authors · 2024-12-03
**Author's Summary of Major Revisions**

Dear All Reviewers,

We sincerely thank all reviewers for their valuable suggestions, which have significantly improved our paper. We provide a summary of the revisions to facilitate discussion between the reviewers and the AC (all revisions are highlighted in purple in the [`new manuscript`](https://openreview.net/pdf?id=ZXs3pkmrRG)).

---
- **Presentation:**
    - Based on feedback from reviewers **ya31**, **8DQE**, and **dQ5H**, our revisions include:
        - ① Reorganizing the _Introduction section_ systematically from the perspectives of the problem, ideas, and techniques, and further refining the writing to better clarify our main contributions.
        - ② Revising the _"When" part_ of the introduction regarding test-time training to provide clearer definitions and avoid unnecessary misunderstandings.
        - ③ Adjusting the order of content in the _Preliminaries section_ to improve clarity and enhance the reading experience.
- **Method:**
    - Based on feedback from reviewers **8DQE**, **ya31**, and **4MTP**, our revisions include:
        - ① Clarifying how our PC-Like SCL algorithm preserves the theoretical identifiability of PC while addressing its empirical limitations. _A new Section D_ has been included in the appendix to address readers' concerns.
    - Following feedback from reviewers **4MTP**, **ya31**, and **dQ5H**, our revisions include:
        - ② We discussed and clarified the convergence guarantee of the posterior distribution in our IS-MCMC algorithm and highlighted key issues that need to be addressed during optimization. _A new Section C_ has been included in the appendix to help readers better understand these insights.
- **Minor Detail:**
    - Based on feedback from reviewers **dQ5H** and **ya31**, our revisions include:
        - ① Rewriting unclear or potentially misleading phrases such as "high-quality training data," "neighborhood," "real-world data," and "unfair baselines."
        - ② Organizing and improving the presentation of confusing content, including the algorithm flow, figure legends, and color differentiation.

---
We hope that all these efforts will greatly address the reviewers' concerns. Once again, thank you!

Sincerely,

Authors of submission 3081

---

### Meta-Review · Area_Chair_bYtv · 2024-12-19

**Metareview:**

The reviews are mixed. We have strong support, borderline support, borderline reject and strong reject. The method, by design, has many moving parts, which makes it naturally hard to understand where the actual contribution lies. I am guessing this caused some reviewers to not fully understand and appreciate the novelty. However, some reviewers explicitly said the proposed approach is not novel, and only combines some existing approaches such as JCI and test time learning. Even the extremely supportive reviewer commented that some claims are overdone and the wording and presentation are very confusing and need to be improved. Their review does not fully match their score and I have to use the content of their review more so than the number given.

Based on where the reviews stand after rebuttal, I believe a round of revision will greatly help emphasize the contribution better.

Based on my own reading, I also recommend that the authors cite and perhaps even leverage the more recent interventional causal discovery algorithms instead of only JCI. Specifically, Jaber et al. 2020 "Causal Discovery from Soft Interventions with Unknown Targets: Characterization and Learning" provides an improvement over JCI and could have been used as the backbone algorithm instead of or in addition to JCI.

**Additional Comments On Reviewer Discussion:**

The reviewers' engagement was minimal. Supportive reviewer seems to have answered the wrong paper and was not reachable afterwards. The critical reviewers were not convinced and did not engage deeply with the authors despite what seems to be a very detailed rebuttal.

---

### Decision · Program_Chairs · 2025-01-22

Reject